# Three-Dimensional Tumor Spheroids as a Tool for Reliable Investigation of Combined Gold Nanoparticle and Docetaxel Treatment

**DOI:** 10.3390/cancers13061465

**Published:** 2021-03-23

**Authors:** Kyle Bromma, Abdulaziz Alhussan, Monica Mesa Perez, Perry Howard, Wayne Beckham, Devika B. Chithrani

**Affiliations:** 1Department of Physics and Astronomy, University of Victoria, Victoria, BC V8P 5C2, Canada; kbromma@uvic.ca (K.B.); alhussan@uvic.ca (A.A.); WBeckham@bccancer.bc.ca (W.B.); 2Department of Biochemistry and Microbiology, University of Victoria, Victoria, BC V8P 5C2, Canada; monikm@uvic.ca (M.M.P.); phoward@uvic.ca (P.H.); 3British Columbia Cancer-Victoria, Victoria, BC V8R 6V5, Canada; 4Centre for Advanced Materials and Related Technologies, Department of Chemistry, University of Victoria, Victoria, BC V8P 5C2, Canada; 5Centre for Biomedical Research, Department of Biology, University of Victoria, Victoria, BC V8P 5C2, Canada; 6Department of Medical Sciences, University of Victoria, Victoria, BC V8P 5C2, Canada; 7Department of Computer Science, Mathematics, Physics and Statistics, Okanagan Campus, University of British Columbia, Kelowna, BC V1V 1V7, Canada

**Keywords:** gold nanoparticles, multicellular spheroids, monolayer, docetaxel, uptake, cell culture, nanomedicine

## Abstract

**Simple Summary:**

Due to the normal tissue toxicity induced by standard therapeutic options such as radiotherapy and chemotherapy, alternative solutions are being explored. Nanomaterials, such as gold nanoparticles, can help reduce side effects by increasing targeted dose to the tumor as well as act as a drug delivery system. Many nano-based systems are tested in a two-dimensional monolayer in the lab, which is not representative of the complex tumor microenvironment. Towards an accelerated translation to the clinic, use of nanomaterials, like the radio-sensitizing gold nanoparticles, were tested in three-dimensional spheroids, which exhibit many properties present in a real tumor such as the extracellular matrix, necrotic cores, and nutrient gradients. This work paves the way for more accurate experimentation in the lab that is more indicative of a real tumor response, leading to a higher chance of success in future clinical studies and clinical treatment.

**Abstract:**

Radiotherapy and chemotherapy are the gold standard for treating patients with cancer in the clinic but, despite modern advances, are limited by normal tissue toxicity. The use of nanomaterials, such as gold nanoparticles (GNPs), to improve radiosensitivity and act as drug delivery systems can mitigate toxicity while increasing deposited tumor dose. To expedite a quicker clinical translation, three-dimensional (3D) tumor spheroid models that can better approximate the tumor environment compared to a two-dimensional (2D) monolayer model have been used. We tested the uptake of 15 nm GNPs and 50 nm GNPs on a monolayer and on spheroids of two cancer cell lines, CAL-27 and HeLa, to evaluate the differences between a 2D and 3D model in similar conditions. The anticancer drug docetaxel (DTX) which can act as a radiosensitizer, was also utilized, informing future potential of GNP-mediated combined therapeutics. In the 2D monolayer model, the addition of DTX induced a small, non-significant increase of uptake of GNPs of between 13% and 24%, while in the 3D spheroid model, DTX increased uptake by between 47% and 186%, with CAL-27 having a much larger increase relative to HeLa. Further, the depth of penetration of 15 nm GNPs over 50 nm GNPs increased by 33% for CAL-27 spheroids and 17% for HeLa spheroids. These results highlight the necessity to optimize GNP treatment conditions in a more realistic tumor-life environment. A 3D spheroid model can capture important details, such as different packing densities from different cancer cell lines, which are absent from a simple 2D monolayer model.

## 1. Introduction

Cancer, a family of diseases arising from dysregulation of genes, is a widespread health issue, with approximately 4950 new cases occurring in the United States per day [1]. The main modalities employed in treatment of cancer, aside from, or in combination with, surgery, are radiotherapy and chemotherapy. Ionizing radiation from radiotherapy can damage cells directly via fragmentation of DNA, such as double strand breaks, or indirectly, via production of free radicals, which accounts for approximately 70% of all damage from high energy X-rays [2,3]. It is estimated that around 50% of patients will require the use of localized radiotherapy throughout their treatment course [4,5]. Likewise, chemotherapy involves the use of anti-cancer drugs acting systemically to accomplish various goals, depending on the type of the cancer [6]. While anti-cancer drugs can be curative in select cancers, chemotherapy is a modality of treatment that is predominately used in adjuvant treatment with radiotherapy and/or surgery to reduce micro-metastases and cancer dissemination [7,8,9]. Both modalities are however constrained in their efficacy due to the limitations of normal tissue toxicity [10]. Further improvements to the clinical response are only of value if the differential response between the tumor demise and normal tissue protection, called the therapeutic index, is increased.

Towards this end, the advent of nanomaterials has allowed for the development of radiosensitizers and drug delivery systems [11,12]. High atomic number noble metal nanoparticles (NPs), such as gold nanoparticles (GNPs; Z = 79) offer many advantages as a model nanomaterial-based system, such as bio-inertness and biostability, while also being simple to functionalize, proving beneficial in an in vivo environment [13,14]. GNPs have previously been shown to act as an effective radiosensitizer in clinically relevant energy ranges [15,16,17]. Targeting of GNPs to tumors can selectively increase deposited dose, increasing the therapeutic index [18]. Moreover, GNPs have an easily modifiable surface, allowing for production of drug delivery systems, many of which have been shown to be effective in the lab as well as in clinical trials [19,20]. Different forms of GNPs can also be synthesized, such as spherical, triangular, stars, nanorods, cages, and more, which will alter the cellular response pathways such as uptake, but can yield more flexibility in designing specific treatment systems [21]. For example, gold nanospheres are often utilized for radiotherapy systems, while gold nanorods and gold nanocages have applicability in photothermal therapy due to their favorable absorption cross-sections [2,22,23]. The use of GNPs as a drug delivery system can improve the pharmacokinetics, the pharmacodynamics, and the biodistribution of various drugs, as well as allow for improved targeting to reduce widespread normal tissue toxicity due to non-specific cytotoxic drugs [24,25]. Combination of two of these applications can lead to the use of GNPs in chemoradiotherapy, a modality shown to be superior to the use of either radiotherapy or chemotherapy alone [26].

Any improvements to any of the previous treatment modalities are dependent on optimization of the transport and uptake of the GNPs into the tumor cells. GNPs enter the cell largely via receptor-mediated endocytosis [27,28]. The uptake of the GNPs on a single cell level depends on many different factors, including size, shape, and surface functionalization [29,30]. However, in an in vivo environment, added complexities of the tumor microenvironment and the immune system create further obstacles to clinical translation [31]. Use of nanomaterials must also consider the interactions that occur in vivo and the journey of the NPs throughout the body. After administration, NPs tend to be excreted in the liver, spleen, and blood, with the clearance half-life depending on many factors, such as size, shape, and surface functionalization [32]. For example, smaller GNPs (<30 nm) would accumulate in the liver and spleen while larger (60 nm GNPs) accumulate in the blood [33]. Without functionalization, bare NPs will form a corona due to opsonin proteins in the plasma, lowering uptake efficiency while increasing removal from the circulatory system by macrophages [26]. Some of these challenges have been tackled; for example, GNPs functionalized with polyethylene glycol (PEG) molecules have shown the ability to evade the immune system and remain in the blood undetected by macrophages [34]. It is important to optimize the properties present in the GNPs. Depending on the size, shape, and concentration, there may be induced toxicity [32]. For example, smaller spherical GNPs (<5 nm) have been found to generate oxidative stress and induce mitochondrial damage, while gold nanorods can also interact with the mitochondria, disrupting the electron transport chain [35,36]. However, a large portion of preclinical work involving GNPs have utilized a two-dimensional (2D) monolayer cell model, which cannot accurately represent the complex heterogenous environment present in the tumor microenvironment. Thus, the introduction of a three-dimensional (3D) model that can more accurately characterize an in vivo environment is essential to more successful clinical translation and to better optimize GNPs in a reliable manner.

The utilization of a 3D spheroid cell model can allow for modelling of an essential aspect of the tumor microenvironment which was previously inaccessible to 2D monolayer models: the extracellular matrix (ECM). The ECM acts as a scaffold for cell growth and is comprised of a network of macromolecules secreted by local cells, mainly polysaccharide glycosaminoglycans, adhesive proteins such as fibronectin, and structural proteins such as collagen [37]. The ECM is also involved in the regulation of cellular function in the tissue, such as cell-to-cell communication, proliferation, and cell adhesion [38]. NPs have also been shown to interact with the ECM via hydrodynamic interactions, physical interactions such as collision of the NPs with the matrix fibers, as well as electrostatic interactions when dealing with charged particles, all of which act as limiting forces to the diffusion of the NPs [39,40]. Functionalization of NPs with suitable peptides to facilitate transport through the ECM can enhance therapeutic effects [39]. Thus, improvements in treatment in an environment that appropriately models these more complex interactions is critical. An important aspect that must be taken into account is the size of the GNPs, which is a characteristic that controls the in vivo blood circulation, tumor accumulation and penetration, and cellular uptake [41]. Due to the inter-matrix spacing present in the ECM, the size of the NPs used is important as the 20–40 nm spacing between collagen fibrils sets the upper limit for effective penetration [42]. The use of spheroids with GNPs has been explored previously in a limited fashion. An exploration of gold nanorods, nanospheres, and oligonucleotide-functionalized nanospheres was undertaken by Rane et al. who showed the difference in uptake profiles present in a 3D model compared to the 2D monolayer [43]. Further studies on how functionalization and size of the GNPs have been explored recently, all agreeing that smaller GNPs will more effectively penetrate deeper into the cells [44,45]. More complex in vivo systems can also be reproduced in a spheroid model, such as a blood-brain barrier made up multiple cell types in a multicellular spheroid. This was explored by Sokolova et al. that showed that ultrasmall GNPs can cross this barrier, illustrating the useful nature of the 3D spheroid [46]. Ultrasmall GNPs have previously been explored in a 3D tumor spheroid model, demonstrating that smaller 2 nm and 6 nm GNPs can penetrate deeper into the tissue when compared to that of 15 nm GNPs [47]. The smaller GNPs also accumulated in the nucleus, due to the nuclear pore being 9 nm in diameter, which could add therapeutics that target the DNA [48]. However, smaller nanoparticles (<10 nm) will have a rapid elimination via renal pathways compared to larger nanoparticles [49]. Thus, a balance between optimized uptake and circulation time due to choice of GNP size is essential. This can be seen in Figure 1, where, in the 2D monolayer, the cells have easy access to the GNPs in serum, removing additional variables that may affect optimization of GNPs at a single cell level. With increased complexity in the 3D tumor spheroid, we can see the introduction of a concentration gradient that will alter the ability of deeper-set cells, which often are hypoxic and thus more treatment-resistant, to uptake GNPs [50]. Optimization in a more complex environment is essential to bridge the gap between the lab and the clinic, to facilitate a quicker translation of the radiosensitizer and drug delivery system platform that GNPs provide.

In this study, we explore the differential uptake due to the alteration of size from smaller 15 nm spherical GNPs to larger 50 nm spherical GNPs in a 2D monolayer model compared to a 3D spheroid model. The GNPs used in this experiment are functionalized with PEG, for stabilization, and a peptide containing integrin binding domain RGD, for tumor targeting, to be suitable for future in vivo experiments. Further, the use of chemotherapeutic agents, such as docetaxel (DTX), in low doses has been shown to significantly alter the uptake, distribution, and retention of GNPs [51]. DTX can also act as a radiosensitizer, due to synchronizing cells in the G2/M phase of the cell cycle [52]. Thus, we will use DTX at low doses, so as not to cause increases in cytotoxicity, in both 2D and 3D models to explore optimum treatment conditions. Exploration of 3D models such as spheroids should yield increased, more accurate throughput of experimental results, hastening the use of important platforms such as GNPs into the clinic.

## 2. Materials and Methods

### 2.1. Synthesis, Surface Modification, and Characterization of Gold Nanoparticles

Gold NPs of size 17 nm and 46nm were synthesized using the citrate reduction method [53]. This was accomplished by adding 300 μL of 1% ${\mathrm{HAuCl}}_{4}\cdot 3H_{2}O$ (Sigma-Aldrich, St. Louis, MO, USA) to 30 mL of double–distilled water and heated on a hot plate while stirring vigorously. Once it reached the boiling point, 300 μL for the larger GNPs, and 1000 μL for the smaller GNPs, of 1% sodium citrate tribasic dihydrate $(\mathrm{HOC}\left(\mathrm{COONa}\right){\left({\mathrm{CH}}_{2}\mathrm{COONa}\right)}_{2}\cdot 2H_{2}O;$ Sigma-Aldrich) was added and mixed. Once the color of the solution changed from dark blue to red, the solution was left to boil for another ten minutes while stirring. Finally, the GNP solution was brought to room temperature while stirring.

The GNPs were PEGylated using PEG of size 2000 Da, along with an RGD peptide of size 1600 Da. PEG was stirred into the GNP solutions such that the grafting density will be 1 PEG molecule per 1 ${\mathrm{nm}}^{2}$ of surface area. For 17 nm GNPs and 46 nm GNPs, this results in 907 PEG and 6647 PEG molecules per GNP to be added to the solution, respectively. Following PEGylation of GNPs, the peptide containing integrin binding domain RGD (CKKKKKKGGRGDMFG) was added to PEGylated GNPs at a ratio of 1 RGD molecule per 2 PEG molecules, referred through this paper as GNP-RGD.

GNPs were characterized using ultraviolet-visible spectrometry (λ Spectrophotometer, Perkin Elmer, Waltham, MA, USA) for approximate size and concentration estimates. Further, dynamic light scattering (DLS) and ζ-potential (LiteSizer 500, Anton Paar, Graz, Austria) were utilized to measure the hydrodynamic diameter and the surface charge of the particles. Stability of the GNPs was measured in phosphate buffered saline (PBS). Imaging of the GNPs with scanning electron microscopy (SEM; SU9000, Hitachi, Chiyoda City, Tokyo, Japan) was used to verify the diameter of the GNPs.

### 2.2. Cell Culture and Growth of Spheroids

HeLa, a human cervical cancer cell line, and CAL-27, a human head and neck cancer cell line, were purchased from the American Type Culture Centre (ATCC, Manassas, VA, USA) and have catalogue numbers CCL-2 and CRL-2095 respectively. Both cell lines were cultured in Dulbecco’s Modified Eagle Medium (DMEM; Gibco, Thermo Fisher Scientific, Waltham, MA, USA) supplemented with 10% fetal bovine serum (Gibco), 1% penicillin and streptomycin (Gibco), and 4 mM GlutaMax (Gibco). For cell dissociation, 0.2% trypysin-EDTA (Gibco) was used.

For 2D and 3D cell models, all cells were initially split from a monolayer of cells at approximately 80% confluency. For the 2D monolayer cell model, cells were plated such that final confluency at the end point is approximately 70% in either six-well plates or 96-well plates, with initial cell count dependent on time and plate type. Once plated, cells are left in the incubator 37 °C and 5% CO_2_ for 24 h to ensure adherence, after which experiments are initiated.

For 3D spheroid cell models, cells are plated in ultra low attachment 96-well microplates (Corning, Corning, NY, USA), with 3125 cells for HeLa per well and 12,500 CAL-27 cells per well. The cells are then centrifuged at 350× *g* for 5 min and left in the incubator at 37 °C and 5% CO_2_. Experiments are initiated once the spheroids form, after incubation for approximately 72 h.

### 2.3. Proliferation Assay of Docetaxel in Monolayer and Spheroids

For the 2D monolayer cell model, 10,000 of each cell line were plated in 96 well black clear bottom microplates (Greiner, Monroe, NC, USA), leaving one column empty for control. Once adhered, each column (8 wells) was treated with a unique docetaxel (DTX) dose ranging from 250 nM to 1.27 $\times 10^{-2}$ nM in media, ensuring that the concentration of dimethyl sulfoxide, which the DTX is stored in, is constant in all columns at approximately 0.05%. Similarly, for the 3D spheroid cell model, once formed, each column (8 wells) was treated with a unique DTX dose ranging from 5000 nM down to 0.254 nM. This was completed in triplicate, on three different plates per cell line.

For monolayer, after 24 h, the media was removed from each sample, cleaned twice with phosphate buffered saline (PBS; Invitrogen, Carlsbad, CA, USA), and left to incubate for a further 48 h in fresh media. For spheroids, after 24 h, half of the media was removed carefully, ensuring the sample was not removed, and then rinsed 5 times with PBS, ensuring approximately a 162× dilution from the initial concentration. Finally, the media was replaced and left for a further 48 h in the incubator.

After 48 h, all the media for the 2D monolayer model was removed and treated with CellTiter-Glo (Promega, Madison, WI, USA) Cell Viability assay, following the manufacturers protocols. Plates were gently mixed for 2 min and luminescence was measured after a further 10 min using a Cytation 1 Cell Imaging Multi-Mode Reader (BioTek, Winooski, VT, USA). Similarly, for the 3D spheroid model, the majority of media was removed, ensuring that the sample was not removed, and treated with CellTiter-Glo 3D Cell Viability assay, following the manufacturer’s protocols. Plates were mixed for 5 min, and luminescence was measured after a further 25 min using the Cytation 1.

### 2.4. Cellular Uptake of Gold Nanoparticle Complex

Incubation with GNPs was done at a concentration of 7.5 μg/mL for all the stated experiments and for all sizes of GNPs. To test the effect that DTX had on the uptake of the GNPs, the different cell models and cell lines were dosed with the desired concentration found in the cell proliferation experiments concurrently with GNP dosing. This correlated to a dose of 20 nM for a monolayer of HeLa and 50 nM for a spheroid of Hela, and a dose of 5 nM for a monolayer of CAL-27 and 200 nM for a spheroid of CAL-27. Samples were incubated at incubator 37 °C and 5% CO_2_ for 24 h following treatment. Cells were then rinsed with PBS at least three times, ensuring no sample is lost when cleaning spheroids, and trypsinized. For spheroid samples, the cells were left in trypsin for 30 min at 37 °C to improve dissociation and produce a single-cell homogenous solution, while the monolayer samples were left in trypsin for approximately ~5 min. The cells were counted using a Coulter Counter (Z2 Coulter; Beckman Coulter, Brea, CA, USA) for GNP quantification per cell.

To measure the gold content for each sample, 500 μL of each sample were treated with 250 μL aqua regia (3:1 ratio of HCl:HNO_3_ (VWR, Radnor, PA, USA) in a 90 °C mineral oil bath for a minimum 30 min. After, 100 μL of hydrogen peroxide (VWR) was added and the samples were returned to the oil bath for 30 min, ensuring complete cell breakdown. These samples were then diluted down to 2.5% *v*/*v* acid content in deionized water and the gold content was quantified using inductively coupled plasma mass spectrometry (ICP-MS; 8800 Triple Quadrupole, Agilent, Santa Clara, CA, USA). Calculation of nanoparticle content based on absolute gold content can be seen in previous publications [54].

### 2.5. Preparation of Cells for Imaging Using Darkfield and Hyper Spectral Imaging

To prepare cells for darkfield imaging for 2D monolayer samples, all cell lines were plated in a six-well plate with glass coverslips placed on the bottom of each well. The cells were then treated as described previously using GNP-RGD and docetaxel at the desired doses. Following treatment, the cells were rinsed three times with PBS and fixed using 4% paraformaldehyde for 20 min at 37 °C. The cover slips were then removed from each well and mounted to a glass slide using Permount Mounting Medium (Fisher Scientific, Waltham, MA, USA).

Similarly, for 3D spheroid samples, following treatment of spheroids with GNP-RGD and docetaxel, the cells were cleaned 5× to dilute GNPs down dramatically, fixed in paraformaldehyde for 30 min at 37 °C, and then embedded in optical cutting temperature (OCT) compound (Fisher) and left in a −20 °C freezer. The spheroids were then sectioned using a MicroTome in a CryoStat (Leica, Wetzlar, Germany) into 10 μM thick sections and adhered to a charged microscope slide. The sections are washed with acetone to remove excess OCT, then rinsed with water. The slides are then stained again with the eosin stain for 30 s and cleaned three times with 100% ethanol. Finally, the cells are rinsed three times with xylenes which acts as a clearing agent. Coverslips were mounted onto the glass slides using Permount Mounting Medium and were dried overnight for microscopy. Each sample, either monolayer or spheroid, was imaged using darkfield microscopy and hyper spectral imaging (HSI; CytoViva, Auburn, AL, USA) under a 10× and 60× objective. The depth of the GNPs from the surface of each spheroid section sample was measured using ImageJ. A minimum of 100 measurements of depth were taken for each sample using the spheroids imaged with the 10× objective.

### 2.6. Cell Cycle Analysis of Docetaxel Using Flow Cytometry

Docetaxel was administered to the cells at desired concentrations as found in proliferation assay. After set time points of 0, 1, 4, 8, and 24 h, the cells were harvested as described previously using trypsin, and a single cell suspension was formed. Cells were washed with PBS and centrifuged at 300× *g* for 5 min twice. Cell pellet was then re-suspended in 1% paraformaldehyde in PBS for fixation and incubated on ice for 15 min. Cells were again washed in PBS and centrifuged at 350× *g* for 5 min. Cells were re-suspended in 0.3 mL PBS and 0.7 mL freezer cold 100% ethanol (overall 70% ethanol). Samples were incubated in the dark at 4 °C for at least an hour, further fixing and dehydrating the cell sample. Samples were then centrifuged at 350× *g* for 10 min at 20 °C. Cell pellet was re-suspended in 1 mL of 0.5% bovine serum albumin (BSA) in PBS, denoted as PBS/BSA and centrifuged at 350× *g* for 5 min at 20 °C. To permeabilize the cell membrane and degrade RNA, the cell pellet was re-suspended in PBTB (PBS, 0.5% BSA, 0.1% Triton-X 100) followed by an addition of RNaseA at a concentration of 100 ug/mL. Samples were then left to shake at 37 °C for 25 min. For labelling DNA, tubes were covered in foil, propidium iodide (PI) added at a concentration of 10 μg/mL and incubated on a shaker at 4 °C for at least 1 h. The cells were then centrifuged at 350× *g* for 5 min at 20 °C. Finally, we re-suspended PI-stained cells in 1 mL of PBS/BSA and passed the solution through a 50 µM cell strainer before running on a flow cytometer (FACS Calibur, BD Biosciences, Franklin Lakes, NJ, USA). Propidium iodide is highly fluorescent at 488 nm with broad emission centered around 600 nm. The amount of DNA content indicates which phase the cell population is in, and thus how synchronized it is.

### 2.7. Statistical Analysis

A Welch’s independent t-test with Bonferroni correction was performed using the statannot Python package. A *p* value < 0.05 was considered statistically significant. Experiments were repeated three times and the data presented is the average, for all experiments.

## 3. Results and Discussion

The size and surface characteristics of the GNP utilized is important to optimize, to improve treatment efficacy and reduce normal tissue toxicity in patients [55,56]. Thus, smaller nanoparticles (~16 nm) and larger nanoparticles (~50 nm) were synthesized, functionalized, and characterized. The GNPs were functionalized with polyethylene glycol (PEG) and a peptide containing integrin binding domain RGD, referred to as RGD, as seen in Figure 2a. The use of PEG is essential for future in-vivo success, as it improves biostability, biodistribution, and increases the circulation half-life [26]. The optimum capping density for PEG molecules onto GNPs was previously found to be 1 PEG molecule for nm^2^ of surface area, which was adhered to in these experiments [57]. While important, PEG has previously been shown to reduce the uptake of GNPs, and thus RGD, which can improve endocytosis and target tumors effectively, was used to improve uptake [34,58]. Scanning transmission electron microscopy was employed to confirm the size of the different GNPs as seen in Figure 2b,c. From a minimum of 100 measured GNPs of each sample, the smaller GNPs were found to have an average diameter of 15.97±1.72 nm while the larger GNPs have an average diameter of 56.12±8.45 nm; these GNPs will be referred to as 15 nm and 50 nm GNPs in this paper. The ζ-potential after conjugation with PEG and RGD was measured as seen in Figure 2d,e, along with the hydrodynamic diameter using dynamic light scattering (DLS) in Figure 2f,g.

Measuring of the ζ-potential and DLS after conjugation is a useful method of confirming functionalization and stability of the nanoparticles. The measured hydrodynamic diameter of the bare GNPs was 19.77 nm and 53.83 nm, respectively, with a polydispersity of 11.27% and 27.34% and a ζ-potential of −42.70 ± 6.16 mV and −38 ± 1.26 mV, respectively. Following conjugation with PEG and RGD, the diameter increased to 31.00 nm and 59.20 nm with a polydispersity of 19.91% and 24.33% and a ζ-potential of −1.09 ± 0.26 mV and 10.30 ± 0.51 mV, respectively. A small peak in the 50 nm GNPs DLS measurements centered around 7 nm is thought to be excess seeds that form in the synthesis process but never congregate into the larger final product. However, these seeds do not make up a significant volume, as verified using the TEM imaging seen in Figure 2c [59]. This does not occur in the 15 nm GNPs spectrum due to it being more energetically efficient to form due to their smaller size. All values can be seen in Appendix A, where the hydrodynamic diameter of both GNPs increases with the addition of PEG and RGD as expected, while the zeta potential grows more positive. This is due to the replacement of the negatively charged citrate molecules, used as a reducing agent to produce the GNPs, with a neutral PEG molecule and a positively charged RGD molecule. Despite the positive trend of ζ-potential after addition of the RGD peptide, the GNPs are still stable due to the addition of PEG molecules. The stability of the GNPs after conjugation with PEG and RGD was verified at each step as seen in Appendix A using UV-Visible spectrometry measurements, which is an approximate measure of size and concentration [60]. An important aspect of the GNPs following conjugation is their ability to resist aggregation in the presence of a human body fluid-like buffer, such as phosphate buffered saline (PBS) [61]. The stability of the GNPs in PBS was verified using dynamic light scattering as seen in Appendix A. This is an important result when moving towards eventual in vivo experiments.

Prior to experimentation with GNPs, the cell lines used in this experiment—CAL−27, a human head and neck cancer cell line, and HeLa, a cervical cancer cell line—were characterized as spheroids as seen in Appendix A. HeLa, a common cell line used in pre-clinical experiments, is intended as a model cell line, while CAL-27 is a tongue cancer cell line, in which the use of cytotoxic drugs like docetaxel can allow for improved treatment response and the use of high doses of radiation is limited due to proximity of many sensitive organs [62]. It can be seen that the different cell lines have different packing density of cells, which is known to alter the efficacy of various treatments [63,64]. Despite this, a volume of approximately 300–400 μM was chosen as an optimum spheroid size, as larger spheroid sizes (>350 μM) introduce a larger core of necrotic cells and increased number of quiescent cells [65]. This corresponds to approximately 3125 HeLa cells and 12,500 CAL-27 cells, with CAL-27 having a larger packing density. It is well known that the packing density of tumor cells will alter the efficacy and penetration of anticancer drugs as well as small nanoparticles, such as micelles [63,64]. The ability to mimic this in vivo effect in the 3D spheroid model is a large benefit over the 2D monolayer model. The cell lines were then characterized in the presence of the anti-cancer drug docetaxel (DTX), as seen in Appendix A. DTX can alter the uptake of the GNPs and act as a radiosensitizer causing synchronization in the highly radiosensitive G2/M phase of the cell cycle [51,52]. For this, and future, experiments, the use of DTX is intended to not increase cell kill but rather synchronize the cells while maintaining viability. Thus, the growth rate inhibition metric introduced by Hafner et al. was utilized to calculate the optimum concentration of DTX from proliferation assays such that the division time is essentially halted and cytostasis is achieved [66]. Based on this metric, a concentration in monolayer of 5 nM and 20 nM, for CAL-27 and HeLa respectively, and a concentration of 200 nM and 50 nM in spheroids, for CAL-27 and HeLa respectively, was used, for a 24-h exposure. The increased resistance to the anticancer drug is expected, due to the more realistic tumor-like environment introduced by the 3D spheroid model. This has been observed in many different spheroid models with many different drugs, including DTX, and is one of the many reasons that the 3D spheroid is better to test drugs than the 2D monolayer [67,68,69]. Due to the introduction of the ECM, penetration of the drug is reduced [70]. When comparing to clinical dosing, the area under concentration-time curve (AUC_0→24_) for HeLa and CAL-27 are 120 nM⋅h and 480 nM⋅h for monolayer, respectively, and 480 nM⋅h and 4800 nM⋅h for spheroids, respectively. Clinically used doses see a median AUC_0→25_ of 1284 nM⋅h at a dose of 20 mg/m^2^ and a median of 5562 nM⋅h at a dose of 100 mg/m^2^, showing the doses used in this experiment are comparable [71].

To verify the effects of these doses of DTX on the cell lines in both the 2D monolayer model and the 3D spheroid model, the cells were imaged using darkfield imaging as can be seen in the HeLa cells in Figure 3a–d. In order to image the spheroids, 10 μm sections of the spheroids were cut using a microtome and then adhered to microscope slides for microscopy, allowing for visualization of deeper internal mechanisms. The DTX can be seen to induce multinucleated cells in monolayer, which is expected due to the dose delivered. DTX at low doses will still have an observable effect on the cells, but not cause complete cell death, allowing for the exploration of dual combination treatment with GNPs. Further, the structure of spheroids appears to be disrupted, reducing the organization of the extracellular matrix produced by the cells in the spheroid, allowing for possibly easier penetration of GNPs into the core. However, the bulk of the DTX-treated spheroid appears to be morphologically similar to the non-treated samples, allowing for the overall function of the spheroids to be maintained. Improved penetration into the tumor is an important feature of all nanomedicine and is one of the main benefits of using a nanoparticle system. Increased penetration of the GNPs in vivo would allow for deeper set radio- and chemo-resistant cells to have an increased dose and higher probability of cell kill. Hyper spectral profiles, from a hyper spectral image (HSI), of the cells and background were added to each darkfield image as an inset figure in Figure 3a–d, to verify the absence of measured GNPs in the image. The use of HSI to verify GNP presence in cells has been used widely in literature from monolayer to in vivo section analysis [72,73]. HSI allows for the capture and spectral analysis of the reflecting and elastically scattered light. Based on the intensity, distribution, and peak of the spectral information collected within each region of interest, we can conclude that no GNPs were present in the control samples. This is completed using a spectral angle mapper classification system that classifies the spectra based on a known spectrum, which in this case is that of the GNPs. Further, it has been shown that the most radiosensitive stage of the cell cycle is the G2/M phase, which DTX is effective at synchronizing a cell population into [74]. To verify synchronization at the chosen dose of DTX in HeLa and CAL-27, the cells were measured following treatment as seen in Figure 3e,f. Over 24 h, the cells transition from a normal cell cycle distribution to a largely synchronized population in the G2/M phase, corresponding to 4n DNA content. Results from CAL-27 can be seen in Appendix A. Thus, the dose of DTX chosen is working as intended.

The next step was to ascertain the difference between the 2D and 3D cell models when treated with both DTX and GNPs. Cells were dosed at a clinically relevant dose of 7.5 μg/mL for 24 h with both sizes of GNPs. There has been a reported time-dependent effect on uptake for bare GNPs observed in monolayer, with a maximum reached after approximately 8 h, while PEG coated GNPs of size 25–40 nm can be seen to have a half-life of approximately 22 h in vivo [30,75]. Thus, a time point of 24 h for total treatment time was chosen to ensure sufficient uptake of GNPs. Further, the dose of GNPs chosen is a clinically-achievable concentration that has shown radiosensitization in vivo previously, at approximately 10 μg/mL [76]. Escalation of GNP dose can be explored in the future to improve therapeutic effects, while ensuring no induced toxicity or negative side effects are introduced. In the 2D monolayer cell model, as seen in Figure 4a–c, we can see that following concomitant dosing of GNPs with DTX, there is a non-significant increase in the uptake of the GNPs compared to dosing with GNPs alone. The use of DTX resulted in an increased uptake of 24% and 15% for CAL-27 and HeLa with 15 nm GNPs, respectively, and an increase of 14% and 13% in uptake of 50 nm GNPs in CAL-27 and HeLa, respectively. This is a similar result found in previous experiments involving GNPs and docetaxel [51]. The difference in efficacy can be attributed to a lower DTX dose utilized in the experiments used in this paper. This increase has previously been attributed to a reduced dilution of the GNPs due to inhibition of cell multiplication; however, the use of DTX has further benefits to the use of GNPs during radiotherapy such as closer localization to the nucleus and longer retention. The smaller GNPs have a larger uptake due to the addition of PEG and RGD on the surface altering the uptake efficiency. This has previously been observed, and is attributed to the larger surface curvature exposing the RGD peptides more efficiently to the integrin receptors on the cell’s surface [51]. This suggests that smaller nanoparticles (<20 nm) will be a better choice with targeting agents, such as the RGD peptide, than larger particles (>20 nm), independent of the cell model used. It is also clear that the different cell lines have different uptake patterns, which is consistent between different sizes of GNPs. For GNPs of size 15 nm and 50 nm, there was an increase of 100% and 130% respectively, in HeLa compared to CAL-27. Darkfield images of the monolayer of CAL-27 cells can be seen in Figure 4d–g, along with an inlay of the spectrum associated with GNPs and with background cells. The inset spectral profile taken from HSI images verifies the presence of GNPs in the samples as measured in the region of interest and classified using a spectral angle mapper. Comparing this spectral data to that of our UV-Visible spectrometry data in Appendix A, there is an observed difference in the peak wavelength measured. This is a result of the intracellular agglomeration in endosomes and lysosomes, causing the GNPs to be in very close proximity to each other. As a result, there will be electromagnetic coupling of the plasmon spectra, leading to a peak at longer wavelengths [77]. This effect will be more exaggerated with the smaller 15 nm GNPs due to a larger number of GNPs being encapsulated in the cells compared to that of 50 nm GNPs, which is what we observe. Visually, the images agree with uptake results, with very similar number of GNPs in the cells, and is consistent with previous results. The ability to see the GNPs with darkfield is due to the aggregation of GNPs inside of endosomes within the cell [55,58]. Similar studies have also shown that larger GNPs (>9 nm) are not likely to be localized in the nucleus unless conjugated with nuclear targeting agents, and thus are likely all within endosomes and lysosomes within the cell [78]. Images of HeLa in monolayer can be seen in Appendix A. Moving into 3D spheroids, we expect an altered uptake response due to the addition of an extracellular matrix.

Spheroids were grown to a size of ~300–400 μm as previously described, as seen in Figure 5a, and then treated with the determined doses of DTX along with 7.5 μg/mL of GNPs for 24 h. Results can be seen in Figure 5b,c, along with darkfield images of 10 μM sections of the CAL-27 spheroids in Figure 5d,g. Similar to the monolayer model, we see larger uptake of smaller 15 nm GNPs with PEG and RGD into the spheroids when compared to the larger 50 nm GNPs, as expected. However, the magnitude of the difference changed, with HeLa only having a 23% increase in 15 nm GNP uptake relative to CAL-27, compared to 100% increase in monolayer. Interestingly, the larger 50 nm GNP uptake still had a 130% increase comparing HeLa to CAL-27. This model dependent behavior is important to note as another advantage when using a 3D spheroid over a 2D monolayer. With DTX, in the case of the 3D spheroid model, for CAL 27, there was a significant (*p* = 0.0016) increase in 15 nm GNP uptake with DTX of 117% and a significant (*p* = 0.012) increase in 50 nm uptake with DTX of 186%, as compared to GNP alone. Likewise, for HeLa in a 3D spheroid model, there was a significant increase (*p* = 0.01428) in 15 nm GNP uptake with DTX of 47% and a significant (*p* = 0.0037) increase in 50 nm GNP uptake with DTX of 56%. This is in contrast with the results seen in the 2D monolayer model; despite controlling for growth rate, there is a dramatic increase in GNP uptake with DTX, with the magnitude depending on the cell line. This may be a result of the reduced organization of the extracellular matrix due to the DTX, as can be seen in the darkfield images, allowing for deeper penetration of GNPs. However, due to the constraints of darkfield imaging, GNPs cannot be properly seen unless aggregated inside of endosomes and lysosomes within the cells. As the number of GNPs that penetrate falls off quickly with depth, the limitations of darkfield imaging hinder confirmation of this theory due to less GNPs being encapsulated in lysosomes. Improved methods of imaging may include confocal imaging and transmission electron microscopy to better analyze the mechanics of encapsulation of GNPs. Another possibility that is also likely is that due to the higher dose of DTX given, cells on the outer radius of the spheroid see a larger effect, which are the same cells that uptake the bulk the GNPs. As a result, the effect seen in monolayer is amplified, due to the larger dose, and we see more GNPs. Images of HeLa spheroids can be seen in Appendix A.

As previously mentioned, the diffusion of GNPs into the spheroids is an important factor when optimizing a nanoplatform, as improved penetration allows for the GNPs to reach the hypoxic and necrotic cores that exist in highly therapeutic-resistant tumors [50]. To see if smaller GNPs had any benefit over larger 50 nm on penetration depth, darkfield images of the sections of spheroids using a 10× objective, as seen in Figure 6a, were analyzed for depth of visible GNPs around the surface. The results can be seen in Figure 6b, where there is a significant (*p* = 0.00057) increase of penetration of 15 nm GNPs over 50 nm GNPs in CAL-27 from 8.83 nm depth for the larger GNPs up to 11.75 nm depth for the smaller GNPs, or a 33% increase. Likewise, there was a significant (*p* = 0.00029) penetration of 15 nm GNPs over 50 nm GNPs in HeLa from 18.59 nm depth for the larger GNPs up to 21.72 nm depth for the smaller GNPs, or a 17% increase. The difference between the two cell lines in actual penetration depth is likely due to the packing density described previously, suggesting that the effectiveness of smaller GNPs over larger GNPs will be more pronounced in tightly packed tumors. The results are in agreement with literature, where smaller nanoparticles tend to penetrate deeper into the 3D spheroid tumors, while the larger nanoparticle aggregate near the surface [79].When moving to more personalized medicine, which is a direction that spheroids can greatly be of benefit, the utilization of patient-derived cells for producing spheroids can allow for in vitro simulation of individual tumor characteristics, such as emulating packing density, something that a 2D monolayer model cannot achieve [80]. Towards analyzing the root cause of the increased uptake of GNPs when using DTX in a spheroid model, the depth of the GNPs was measured similar to above, with the results in Figure 6c,d. There was a slight, non-significant increase in the penetration depth for most samples, except for HeLa using 15 nm GNPs. Thus, the benefits of DTX do not significantly extend to the penetration depth of the gold nanoparticles.

Though the number of smaller 15 nm GNPs captured by the cells was larger than the number of larger 50 nm GNPs in both cell lines and both cell models by an order of magnitude, the total gold content, measured in picograms of Au per cell, was larger for the 50 nm GNPs, as seen in Appendix A, due to the larger total volume of the 50 nm GNPs. There was an average of 157%, 30%, 238%, and 106% increase in total Au content when moving from 15 nm to 50 nm GNPs, for a monolayer of CAL-27, CAL-27 spheroids, HeLa in monolayer, and HeLa spheroids, respectively. An important finding that must be elucidated in future studies is if the gold content of the larger GNPs outweighs the increase in nanoparticle content due the smaller GNPs following treatment with radiation therapy, in the 3D model vs. a 2D model.

There are many differences when it comes to the environments present in a 2D vs. a 3D model. A spheroid-based 3D model introduces the extracellular matrix and enables optimizing of nano-based systems due to more realistic, tumor-like conditions. This study suggests that GNPs of different sizes will have different uptake effects based not only on the cell model—2D vs. 3D—but also on the cell line, highlighting a key necessity to use 3D models to properly simulate treatments in vitro before transition to an in vivo environment. However, the benefits of the 3D spheroid model in modeling radiation combined with GNPs still must be explored in future studies. Further, biochemical analyses of the effect that DTX has on the 3D spheroids when combined with GNPs should be elucidated, beyond just the altered uptake response.

## 4. Conclusions

Gold nanoparticles present an exciting method to improve tumor response while reducing normal tissue toxicity. While shown to be very effective in vitro as a radiosensitizer as well as a drug delivery system, translation to the clinic has been slow [81]. Introduction of a more complex 3D tumor spheroid model that can more accurately represent the complex environment present in vivo, compared to a 2D monolayer, can facilitate a more rapid translation. Towards this goal, we tested small 15 nm GNPs and larger 50 nm GNPs in the two different in vitro models, monolayer and spheroids, and in two different cell lines, CAL-27 and HeLa. The use of DTX, an anticancer drug that can act as a cell cycle synchronizing agent at lower doses to improve uptake, was also explored in both cell models. When moving into the 3D spheroid model, there were clear benefits over the use of the 2D monolayer model, allowing for better approximation of the effects of cell density and the resulting extracellular matrix. While the uptake data followed a similar trend in monolayer and spheroids, the use of DTX had a more pronounced effect on the spheroids, increasing uptake significantly, while the depth of penetration of different sizes of GNPs, an important characteristic that allows for targeting of deep-set tumor cells, found that smaller 15 nm GNPs penetrate more effectively compared to the larger 50 nm GNPs. The use of smart nanomaterials, such as the GNPs used in this study, has great potential due to their ability to act as radiosensitizers and as drug carriers. Our results demonstrate that the 2D preclinical monolayer model present an unrealistic representation of the real tumor-like environment, which could lead to inaccurate screening of the potential of GNPs and other nanomedicines. The proposed 3D spheroid model is, we believe, an essential tool to validate the efficacy of novel therapeutics, due to the ability of these models to mimic many of the key features that are present in a real tumor microenvironment. Therefore, these results showcase the importance in using more accurate in vitro models, such as 3D tumor spheroids, to accelerate the use of nanomedicine into the clinic.

## Figures and Tables

**Figure 1 cancers-13-01465-f001:**
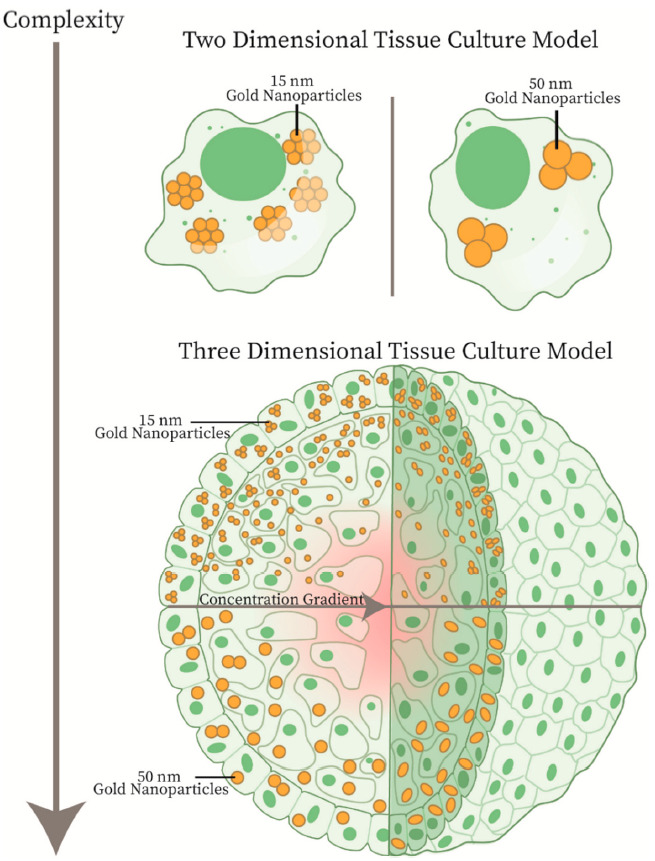
Two-Dimensional Tissue Culture Model vs. Three-Dimensional Tissue Culture Model. In a two-dimensional monolayer model, the gold nanoparticles have easy access to all the cells, allowing for optimization of uptake at a single cell level. However, the complexity of an in vivo environment is better approximated by the three-dimensional spheroid model. In a spheroid model, the uptake of smaller nanoparticles will penetrate deeper, allowing for targeting of difficult to reach cells. Further, a spheroid model allows for optimization in a tumor-like microenvironment, which can be largely affected by concurrent treatment with chemotherapy or radiotherapy.

**Figure 2 cancers-13-01465-f002:**
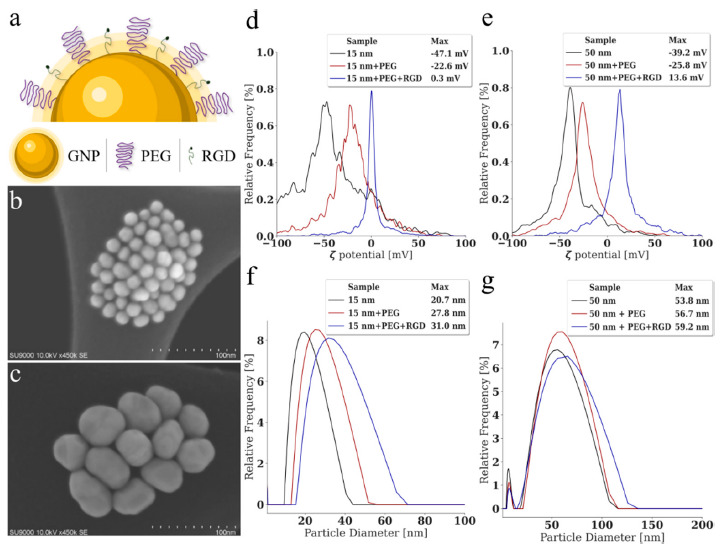
Characterization of gold nanoparticles. (**a**) A schematic diagram of the functionalization of the gold nanoparticles, which are conjugated with polyethylene glycol and a peptide containing integrin binding domain RGD. (**b**,**c**) Transmission electron microscope images of the small and large gold nanoparticles. Scale bar is 100 nm. (**d**,**e**) ζ-potential and (**f**,**g**) dynamic light scattering measurements of the gold nanoparticles prior to and following conjugation with polyethylene glycol and the RGD peptide.

**Figure 3 cancers-13-01465-f003:**
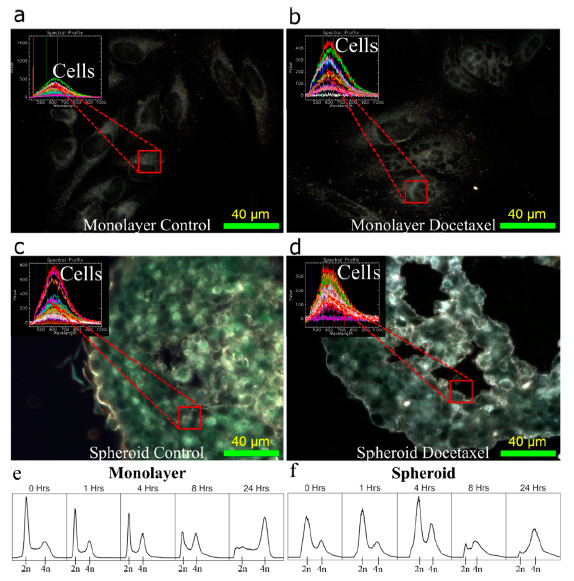
Synchronization of tissue culture models with docetaxel. (**a**,**c**) Darkfield images of a (**a**) monolayer of HeLa cells and a (**c**) HeLa spheroid section. (**b**,**d**) Darkfield images of a (**b**) monolayer of HeLa cells and a (**d**) HeLa spheroid section treated with docetaxel. Overlay: Hyper spectral imaging yields a spectrum that shows no gold in the images. Scale bar is 40 μm. (**e**,**f**) Cell cycle distribution after 0, 1, 4, 8, and 24 h of treatment with docetaxel in a (**e**) monolayer of HeLa and a (**f**) HeLa spheroid.

**Figure 4 cancers-13-01465-f004:**
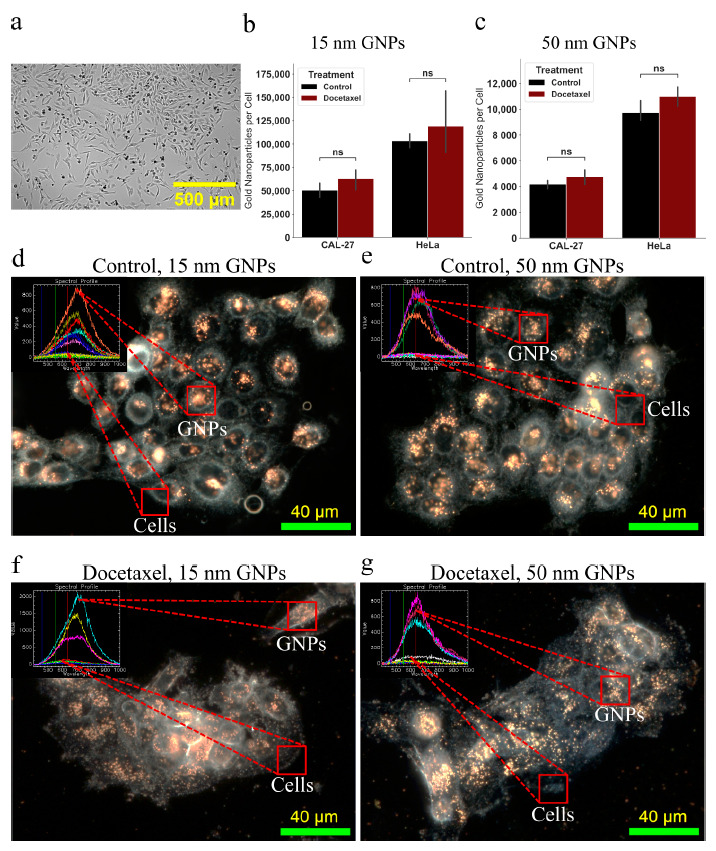
Gold nanoparticle update in two-dimensional cell model. (**a**) Brightfield image of a monolayer of HeLa in a 96-well microplate. Scale bar is 1000 μm. (**b**,**c**) Uptake of the 15 nm and the 50 nm functionalized gold nanoparticles, with and without docetaxel. Treatment time was 24 h. Error bars signify one standard deviation from average of three independent measurements. (**d**–**g**) Darkfield imaging of CAL-27 cells with 15 nm and 50 nm functionalized gold nanoparticles, with and without docetaxel. Overlay: Hyper spectral imaging yields a spectrum that is matched to the gold in the sample. Scale bar is 40 μm. ns indicates no significance.

**Figure 5 cancers-13-01465-f005:**
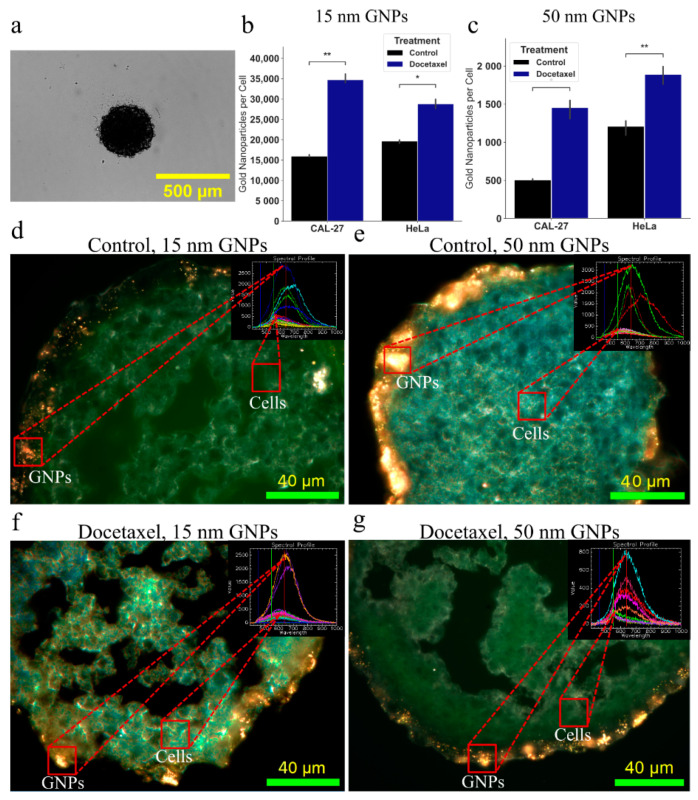
Gold nanoparticle update in three-dimensional cell model. (**a**) Brightfield image of a spheroid of HeLa in a 96-well microplate. Scale bar is 1000 μm. (**b**,**c**) Uptake of the 15 nm and the 50 nm functionalized gold nanoparticles, with and without docetaxel. Treatment time was 24 h. Error bars signify one standard deviation from average of three independent measurements. (**d**–**g**) Darkfield imaging of CAL-27 spheroid with 15 nm and 50 nm functionalized gold nanoparticles, with and without docetaxel. Overlay: Hyper spectral imaging yields a spectrum that is matched to the gold in the sample. Scale bar is 40 μm. * indicates 0.01 < *p* < 0.05, ** indicates 0.001 < *p* < 0.01.

**Figure 6 cancers-13-01465-f006:**
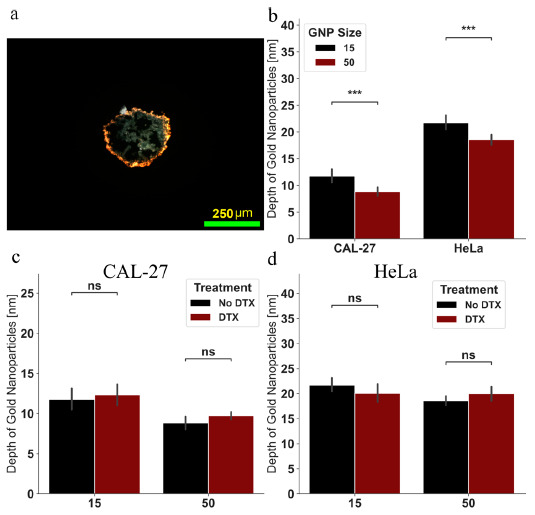
Depth of penetration of gold nanoparticles. (**a**) Darkfield image of HeLa spheroid with 50 nm functionalized gold nanoparticles without docetaxel. Scale bar is 250 μm. (**b**) Depth of visible penetration of gold nanoparticles as measured using darkfield images of spheroids from the 10× objective. (**c**,**d**) Depth of gold nanoparticle visible penetration with and without docetaxel for (**c**) CAL-27 and (**d**) HeLa. Error bars represent 1 standard deviation from a minimum 100 measurements. ns indicates no significance, *** indicates 0.0001 < *p* < 0.001.

## Data Availability

Datasets generated and/or analyzed during the current study are available from the corresponding author on reasonable request.

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
