# Peer review of "Three-Dimensional Tumor Spheroids as a Tool for Reliable Investigation of Combined Gold Nanoparticle and Docetaxel Treatment"

_cancers, 2021, doi:10.3390/cancers13061465_

Round 1

Reviewer 1 Report

This report describes the vital phenomenon in cancer therapy. In this nano-era, deeper insights about the nanoparticles (NPs) applicability should be more explored, making it closer to clinical delivery, as described in this paper. Although this manuscript adds an essential viewpoint in anticancer NPs, it still needs to clarify the following for better readership.

  1. From the stability perspective, are these prepared NPs not aggregating, as it showed the low zeta potential after surface modification (<+15mV). And what indicates the small peak appeared about 5nm scale in the size data of 50nm samples?
  2. How sure is the author about the internalization of NPs? From the images shown, the NPs may adhere to the cells' surface but not inside cell surfaces such as the cytoplasm or nucleus, which may bring a false claim?
  3. Since the applied microscope hinders the exact localization of GNPs except for the visible edges of the spheroids, would creating a 3D image using the microscope software provide deeper understandings?
  4. Besides, I recommend imaging a cross-section of the already fixed spheroid, which will clearly show the deeper internalization of the GNPs, if the author cannot do other specific techniques. Also, the author should describe the insets presented in the microscope images.
  5. Could you please clarify what sort of changes this entire study brings out in terms of TME? Although the author described the differences in NPs internalization (with size difference) between mono 2D and 3D models, still deeper molecular or biochemical analyses would add more exciting values to this work. 
  6. The author should mention PFG and RGD in terms of NPs internalization apart from the anticancer drug. 

Reviewer 2 Report

The manuscript aims at evaluating the use of 3D tumor spheroid models as a more realistic tumor model in the evaluation of the nanoparticle uptake and penetration depth in gold-nanoparticle (GNPs) combined therapeutics with anticancer drug model DTX. The number of published works on nanoparticle uptake by spheroids has been increasing over the last decade, but the GNPs interaction with spheroids are still poorly understood.  Thus, the manuscript is of interest to the readers of this journal. I have looked at this paper from the point of view of a researcher with an academic knowledge of gold nanoparticles and experienced in optical spectroscopy and fluorescence microscopy of labelled 2D cell cultures. I am not  able to emit a well-supported opinion on the procedures for the development of the 3D tumor spheroids. Within my field of expertise, I think that the design of the experiment suffers from a serious limitation of the darkfield microscopy to evaluate the penetration depth of GNPs related with the limited contrast provided by non-aggregated GNPs using this technique. However, the authors duly recognize such limitations in the methodology and still provide a reliable interpretation of the results supported by the data provided.

Overall I feel positive about publication of the manuscript, but I do find a few issues that require major revision:

  1. The revision of the literature addressing the issue of GNP uptake in spheroids is incomplete. The reference to some interesting recent works is missing, such as . doi: 3390/nano10102040, doi: 10.1002/mabi.201900221, doi: 10.1371/journal.pone.0167548 and doi: 10.1021/nn301282m
  2. The critical comparison of the obtained results with literature data is insufficient.
  3. The title of the paper seems to be off target. The title implies that this study addressed the suitability of three-dimensional tumor spheroids as a tool to optimize the nano-bio interface. To make such study one would have to compare the results in the spheroids with those on live animal models, which is not the case. I suggest the authors revise the title to match the manuscript content.
  4. I would like to have more details on the estimation of the actual PEG coverage on the GNPs. Not the intended PEG coverage, but the one that was really obtained. The PEG coverage might also have a significant effect on the NP uptake.
  5. In page 5, the authors mention the use of a constant concentration of DMSO. The value of such concentration should be given.
  6. In page 7,Figure 2, the quality of the SEM image should be improved. The 50 nm GNPs appear to have a high dispersion in shape. This relates to the fact that in fig. S1 the absorption spectra of the 50 nm GNPs have a shape that differ significantly from those typically reported in the literature. From the 15 nm to the 50 nm GNPs we expect to observe only a broadening of the plasmon resonance and a small redshift in the plasmon resonance, and yet the authors report a size effect in the spectral shape that goes well beyond the expected changes. In particular, the absorption on the blue ao the plasmon resonance peak is significantly reduced to a point that it does not look like a plasmon resonance spectra anymore. It seems that not only the size has changed but also the shape of the particles changed significantly. The particle shape should be characterized more thoroughly. It is well known that in addition to size, shape can also affect GNPs uptake.
  7. In Figure 3, it is not clear what is the added value of insets showing a series of spectra. What kind of spectra are those? Are those scattering spectra? The spectra do not appear to change in shape only the peak intensity is changing. It is not clear to which region within the square region of interest (ROI) each spectrum belongs. It should be more informative to show an intensity profile along some relevant line. The spectra should either be adequately discussed in the manuscript or placed in supplementary information.
  8. In figure 4, authors should comment on the fact that the scattering spectra of the GNPs peak at different wavelength from the absorption spectra shown in figure S1. Also unexpected, and equally worth of an explanation, is the fact that in the smaller GNPs the spectra appear shifted to longer wavelengths (centered at 700 nm) when compared to the 50 nm GNPs (centered at 650 nm).  In solution both spectra appear centers at 520-550 nm.

Reviewer 3 Report

In this submission, Bromma et al reports the uptake of 15 nm GNPs and 50 nm GNPs on a monolayer and on spheroids of two cancer cell lines, CAL-27 and HeLa, to evaluate the differences between a 2D and 3D model in similar conditions. This is an interesting study. Although there are a few unclear mechanisms which need to be explained. I recommend this paper to be accepted with subject to major revisions. Following are my specific comments;

  • Line 103 – ‘Another important aspect that must be taken into account is the size of the GNPs, which is a characteristic that controls the in vivo blood circulation, tumor accumulation and penetration, and cellular uptake’. Authors did not comment on clearance and excretion of GNPs. Authors have elegantly described GNPs and their potential therapeutic outcomes. I suggest to add a little bit more details on why off-target toxicity needs to be overcome and how GNPs can induce toxicity, mitochondria respiration, DNA strand break. What are excretion pathways of GNPs? As such the clearance and excretion mechanisms of GNPs is a current area of research activity. Of course it is important to explore this at cellular/subcellular levels to evaluate therapeutic outcome, but it is significantly important to determine where GNPs end up at organ level. What are potential excretion pathways of GNPs. I suggest authors to discuss this around the following paper (https://doi.org/10.1002/advs.201903441).
  • I invite authors to discuss the effects of shape of GNPs on therapeutic outcomes such as nanorods, nanospheres, nanoprisms, etc.
  • The functionalization aspects and their crucial effects on uptakes of GNPs needs to be elaborated, and how ECM enzymes/proteins can avoid corona formation during the targeted delivery of GNPs.
  • Figure 4, 5 - ‘three measurements’, are authors referring to three independent experiments? If yes, please specify.
  • Figure 4 – when authors mention cellular uptake of GNPs, I suggest to add treatment time as well. It has extensively been reported that uptake of GNPs is time dependent. Since the uptake time determines when and how GNPs can be administered to a living system. Can authors shed some light of the time-dependent uptake of GNPs at cellular/subcellular levels.
  • Why did authors choose these cell lines? and how did they choose the concentration ranges and the relevance of such low concentrations to real-world models (in a living system). Why not to compare the efficacy in a concentration and time-dependent manner?
  • I suggest authors not to abbreviate words which they are only using once or twice throughout the manuscript, for instance, DDSs, TME. Although they used drug delivery systems again (line 70) where they did not use the abbreviated term. Please check this carefully.

Round 2

Reviewer 2 Report

The authors have effectively addressed the identified issues and improved the overall quality of the manuscript.

I feel positive about publication of the manuscript in its revised form.

Reviewer 3 Report

I am pleased to recommend the revised manuscript for publication in Cancers,